# High-Pressure Synthesis of Non-Stoichiometric $Li_xWO_3$ ($0.5 \leq x \leq 1.0$) with $LiNbO_3$ Structure

**Kohdai Ishida [1]** , **Yuya Ikeuchi [1]** , **Cédric Tassel [1]** , **Hiroshi Takatsu [1]** , **Craig M. Brown [2,3]** and **Hiroshi Kageyama [1,\*]**

[1]   Department of Energy and Hydrocarbon Chemistry, Graduate School of Engineering, Kyoto University, Nishikyo-ku, Kyoto 615-8510, Japan; ishida.kodai.83u@st.kyoto-u.ac.jp (K.I.); yuyapple@me.com (Y.I.); cedric.tassel@gmail.com (C.T.); takatsu@scl.kyoto-u.ac.jp (H.T.)

[2]   Center for Neutron Research, National Institute of Standards and Technology, Gaithersburg, MD 20899, USA; craig.brown@nist.gov

[3]   Department of Chemical and Biochemical Engineering, University of Delaware, Newark, DE 19716, USA

\*   Correspondence: kage@scl.kyoto-u.ac.jp; Tel.: +81-075-383-2506

**Abstract:** Compounds with the $LiNbO_3$-type structure are important for a variety of applications, such as piezoelectric sensors, while recent attention has been paid to magnetic and electronic properties. However, all the materials reported are stoichiometric. This work reports on the high-pressure synthesis of lithium tungsten bronze $Li_xWO_3$ with the $LiNbO_3$-type structure, with a substantial non-stoichiometry ($0.5 \leq x \leq 1$). $Li_{0.8}WO_3$ exhibit a metallic conductivity. This phase is related to an ambient-pressure perovskite phase ($0 \leq x \leq 0.5$) by the octahedral tilting switching between $a^-a^-a^-$ and $a^+a^+a^+$.

**Keywords:** high pressure; $LiNbO_3$; $LiWO_3$; tungsten bronze

## 1. Introduction

The lack of inversion symmetry in $LiNbO_3$ allows ferroelectricity, piezoelectricity, pyroelectricity, and second-order nonlinear optical behavior [1,2]. These properties of $LiNbO_3$ and related oxide insulators (e.g., $LiTaO_3$ and $ZnSnO_3$) lead to various industrial applications, such as waveguides, modulators, nonlinear crystals, and piezoelectric sensors. Recent research has added magnetic and electronic properties to this structural type. To obtain such materials with finite *d* electrons, the high-pressure synthesis approach was employed. $ScFeO_3$ exhibits a magnetic ordering far above room temperature [3], while in $MnTaO_2N$ exhibits a strong bending of Mn–O–Mn angle, in the extended $MnO_6$ network introduces spin frustration, resulting in a spiral spin order at low temperature [4]. $LiOsO_3$ undergoes a structural phase transition from a centrosymmetric ($R-3c$) structure to a noncentrosymmetric ($R3c$) structure at 140 K [5]. A "polar" metallicity in the low-temperature phase may offer an interesting platform for exotic electronic phases to control carrier density by, e.g., introducing anion/cation deficiency or substitution is essential. Unlike perovskites, however, all the known $LiNbO_3$-type compounds are stoichiometric.

The present work stems from our recent studies on potassium and sodium tungsten bronzes, revealing the capability of high-pressure synthesis to increase the alkali-metal content [6,7]. For $K_xWO_3$, high pressure allows the formation of a stoichiometric tetragonal phase $K_{0.6}WO_3$ ($K_3W_5O_{15}$) with anomalous metallic behavior and $KWO_3$ with the ideal perovskite ($Pm-3m$) structure [6]. The high-pressure methodology also produces a stoichiometric $NaWO_3$, in which a distorted perovskite ($Im-3$) structure gives rise to a novel rattling phenomenon [7]. From these results, it is clear that the high-pressure condition can suppress undesirable volatilization of alkali metals. The high compressibility of alkali metals may also allow for more incorporation of Na or K ions into the available lattice site.

This study targeted lithium tungsten bronze, $Li_xWO_3$, which is known to exist in the compositional range of $x \leq 0.5$ and has the perovskite ($Im-3$) structure, as shown in Figure 1a [8]. This phase has typically been prepared by a conventional high-temperature solid state reaction [9,10] and electrochemical reaction [11,12]. We report that the use of high pressure stabilizes the $LiNbO_3$-type structure (Figure 1b), for the first time in the tungsten bronze family. In addition, $Li_xWO_3$ is *A*-site non-stoichiometric ($0.5 \leq x \leq 1$) with the variable *d* electron count of 0.5~1. We show the structural characterizations by means of X-ray diffraction (XRD) and neutron diffraction (ND), along with physical properties by electrical resistivity measurement.

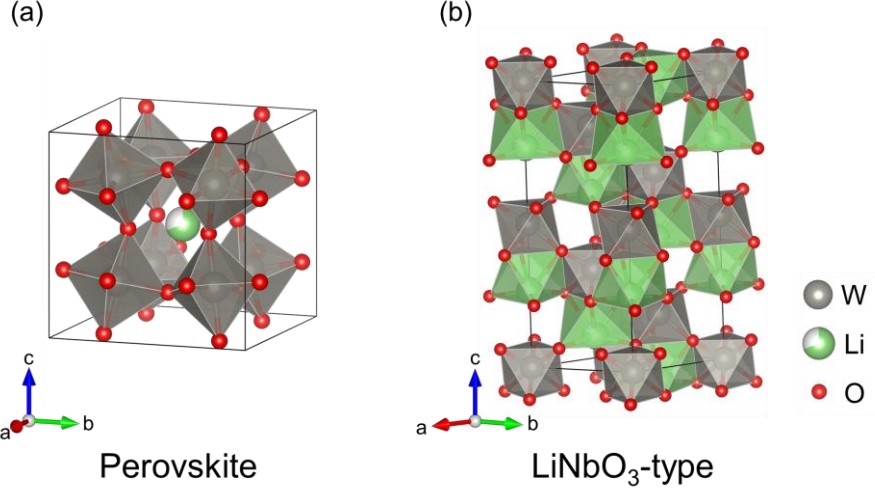

**Figure 1.** Crystal structures of $Li_xWO_3$ for (**a**) $0 \leq x < 0.5$ with the cubic perovskite ($Im-3$) structure reported previously [8] and (**b**) $0.5 \leq x \leq 1$ with the $LiNbO_3$-type ($R3c$) structure obtained in this work. The grey, green and red balls represent W, Li and O atoms, respectively. W- and Li-centered octahedra are shown in grey and green.

## 2. Results and Discussion

### 2.1. Synthesis of $Li_xWO_3$ ($0.5 \leq x \leq 1$)

Figure 2a shows the XRD patterns of $Li_xWO_3$ with $0.5 \leq x \leq 1$. The diffraction patterns for all the samples are markedly different from that of the cubic ($Im-3$) perovskite for $x < 0.5$ [8] but are similar to that of $LiNbO_3$ (Figure 2a). Single phases were obtained for $x = 0.7$, 0.8 and 0.85, while unreacted $Li_2WO_4$, $WO_2$, and $WO_3$ impurities were observed for the remaining samples. Modifying reaction temperature and pressure did not improve the results. Figure 2b shows the lattice parameters, *a* and *c*, as determined from Le Bail analysis. It is seen that the cell parameters evolve anisotropically as a function of *x*. When *x* is increased, the *a* and *c* axes decrease and increase, respectively. The linear evolution of both cell parameters, following the Vegard's law, ensures the successful preparation of a solid solution for $0.5 \leq x \leq 1$, though impurities were present for $x \leq 0.6$ and $x > 0.85$. The difference in the density of the ambient-pressure phase of $Li_{0.5}WO_3$ and its high-pressure polymorph is very small (0.40%). The rhombohedral phase of $Li_xWO_3$ was extremely unstable at ambient temperature. Even in an inert atmosphere (e.g., in an Ar- or $N_2$-flled glovebox), the sample starts to decompose within a few days into the known cubic ($Im-3$) perovskite of $Li_xWO_3$, possibly accompanied by Li extraction. The characterization of the materials was, therefore, carried out immediately after the syntheses, otherwise the samples were stored in liquid nitrogen.

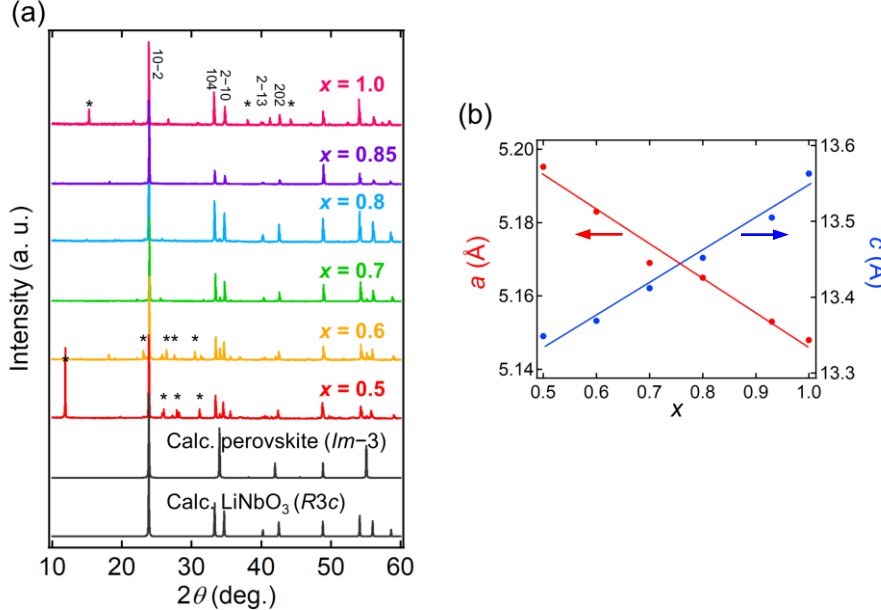

**Figure 2.** (**a**) XRD patterns of $Li_xWO_3$ ($0.5 \leq x \leq 1.0$) synthesized at 5–8 GPa, demonstrating the formation of the rhombohedral structure. Calculated patterns for the perovskite ($Im-3$) and $LiNbO_3$ ($R3c$) structures are shown for comparison. Asterisks denote unreacted starting materials of $Li_2WO_4$, $WO_2$, and $WO_3$. (**b**) Lattice parameters, *a* (red) and *c* (blue), of the rhombohedral $Li_xWO_3$ phase as a function of *x*. The errors are within the size of the symbols. The solid lines are linear fits to the data.

## 2.2. Structural Characterizations of $Li_{0.8}WO_3$

Extinction diffraction peaks in the XRD profiles suggest that the space group of the new phase is $R3c$ or $R-3c$. We performed Rietveld refinement of the XRD data for the phase pure $x = 0.8$, assuming a non-centrosymmetric $LiNbO_3$-type structure ($R3c$), as shown in Figure 3a. Li, W and O were placed at the 12*c* (0, 0, *z*), 6*b* (0, 0, 0) and 18*e* (*x*, *y*, *z*) site, respectively. Any Li deficiency was not taken into consideration at this stage due to low X-ray contrast for Li. The refined parameters are shown in Table 1. In order to examine the Li content, neutron measurements for $x = 0.8$ were conducted using the BT-1 powder diffractometer. Since the sample partially decomposed to the perovskite phase $Li_xWO_3$ ($x < 0.5$) with the $Im-3$ space group during shipping, the refinement was performed with inclusion of cubic-$Li_{0.5}WO_3$ (21.6(8)%) as a secondary phase, which resulted in goodness-of-fit parameters $R_{wp} = 8.95\%$, $R_p = 6.72\%$, and GOF = 1.16 and reasonable atomic parameters (Figure 3b and Table 1). The Li occupancy was estimated as 0.77(5), which is consistent with the nominal content of $x = 0.8$. The refinement of the 5 K data gave $R_{wp} = 9.38\%$, $R_p = 7.13\%$, and GOF = 1.41. We calculated an $AO_6$ octahedral distortion parameter, $\Delta = 1/6\Sigma[(d_i - \langle d \rangle)/\langle d \rangle]^2$, where $d_i$ is the individual bond distance, and $\langle d \rangle$ is the average bond length [13]. We obtained $\Delta = 7.79 \times 10^{-3}$ for $Li_{0.8}WO_3$. This value is larger than other +1/+5 type compounds (e.g., $4.27 \times 10^{-3}$ for $LiTaO_3$ [14] and $1.68 \times 10^{-3}$ for $LiNbO_3$ [15]) but is comparable with $8.15 \times 10^{-3}$ for $LiOsO_3$. The difference might be related to the *d* electron count. Refinement of the same neutron data with a $R-3c$ model also gave similar reliability factors of $R_{wp} = 8.95\%$, $R_p = 6.72\%$ and GOF = 1.16 ($R_{wp} = 9.36\%$, $R_p = 7.11\%$ and GOF = 1.40 at 5 K). As such, we cannot completely rule out the possibility of a high-*T* $LiNbO_3$ form with a centrosymmetric space group of $R-3c$ without suitable single crystals. Note that the observed extinction reflections excluded the ilmenite-type structure ($R-3$).

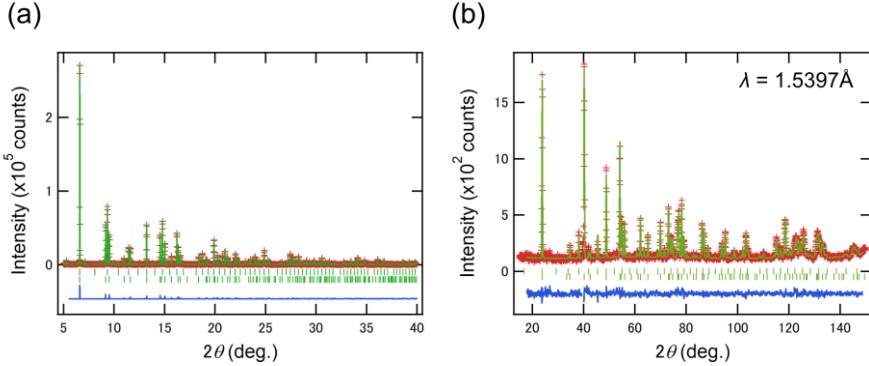

**Figure 3.** Rietveld refinement of (**a**) XRD and (**b**) ND data for $Li_xWO_3$ ($x = 0.8$) assuming the $LiNbO_3$-type ($R3c$) structure. Red crosses, green solid line, and blue solid line represent observed, calculated, and difference intensities, respectively. The top and bottom green ticks indicate the positions of the Bragg peaks of the $LiNbO_3$-type structure and the cubic ($Im-3$) phase with $x \sim 0.5$ (see text for details).

**Table 1.** Structural parameters of $Li_{0.8}WO_3$ from Rietveld refinement on XRD and ND data at 300 K. $g$ is the site occupancy factor. XRD: $a = 5.1626(4)$, c $= 13.4434(2)$ Å, $R_{wp} = 12.43$, $R_p = 9.21\%$ and GOF = 2.24. ND: $a = 5.1665(3)$, c $= 13.4424(10)$ Å, $R_{wp} = 8.95$, $R_p = 6.72\%$ and GOF = 1.16.

| Technique | Atom | Wyckoff Position | $g$ | $x$ | $y$ | $z$ | $U_{iso}$ (Å$^2$) |
|---|---|---|---|---|---|---|---|
| XRD | Li | 12$c$ | 1 | 0 | 0 | 0.2502(6) | 0 |
| | W | 6$b$ | 1 | 0 | 0 | 0 | 0 |
| | O | 18$e$ | 1 | 0.076(3) | 0.371(6) | 0.0829(2) | 0 |
| ND | Li | 12$c$ | 0.77(5) | 0 | 0 | 0.2748(16) | 0.014(6) |
| | W | 6$b$ | 1 | 0 | 0 | 0 | 0.0034(8) |
| | O | 18$e$ | 1 | 0.0659(12) | 0.332(2) | 0.0821(14) | 0.0063(5) |

### 2.3. Structural Transition in $Li_xWO_3$

Together with past research, this study has demonstrated the occurrence of compositional transition from the perovskite ($x \leq 0.5$) to the $LiNbO_3$-type structures ($x \geq 0.5$). Given that the latter structure has a network composed of corner-sharing $WO_6$ octahedra, one can then discuss this structural transition within the framework of perovskite chemistry. Using the Glazer notation, the observed compositional transition can be viewed as an octahedral tilting switching from three in-phase rotations ($a^+a^+a^+$) for $x \leq 0.5$ to three out-of-phase rotations ($a^-a^-a^-$) for $x \geq 0.5$, a transition which has not been observed in any perovskite-based materials. It is interesting to compare this system with the sodium, $Na_xWO_3$, where the $a^+a^+a^+$ structure ($Im-3$) is stable over higher Na concentration up to the full stoichiometry ($0.8 \leq x \leq 1$) [7]. Our recent study on $Na_xWO_3$ revealed an unusual local phonon dynamic, which is interpreted as a rattling phenomenon based on loosely bound Na atoms at the 6$b$ site with 12-fold coordination. This observation implies that the replacement of Na by smaller Li cations destabilizes the $a^+a^+a^+$ structure. $Li^+$ ions would be well suited to the six-fold coordination in the $a^-a^-a^-$ structure ($x \geq 0.5$).

### 2.4. Physical Properties

The $LiNbO_3$-type compounds are already successful in industrial applications, such as piezoelectric sensors and optical modulators, but the recent discovery of $LiOsO_3$ opened a new avenue for the study of a "polar" metal, provoking many discussions on its mechanism [16–18]. What differentiates $Li_xWO_3$ ($0.5 \leq x \leq 1$) from $LiOsO_3$ ($d^3$) and $MnTaO_2N$ ($d^5$) is the variable $d$ electron count from 0.5 to 1.0, apart from a trace amount of defects in $LiNbO_3$ (12$c$ site) [19], thus giving a unique opportunity to tune and understand the physical properties of the $LiNbO_3$-type structure. Unfortunately, the unstable nature of the samples and the presence of the impurity phases did not permit a systematic study. For this

reason, we show hereafter the temperature dependence of the electrical resistivity ($\rho$) of $Li_{0.8}WO_3$, where two batches of samples (#1, #2) were used. As shown in Figure 4, both specimens show a metallic behavior in the low-temperature region below 40 K. However, we found a slightly negative temperature dependence ($d\rho/dT < 0$) above 40 K, which might be intrinsic but could be due to the effect of some grain boundary impedance. In addition, there is a drop below 3 K, which could be indicative of superconductivity, but the strong sample dependence and the absence of a peak in heat capacity measurements (not shown) indicate that this more likely arises from an impurity.

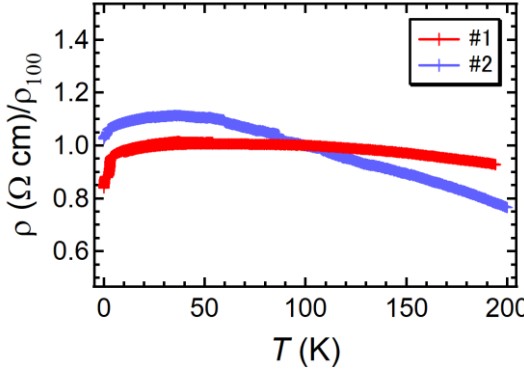

**Figure 4.** The temperature dependence of the electric resistivity of $Li_{0.8}WO_3$ normalized for the 100 K value. #1 and #2 are the batch numbers. $\rho_{100}$ = 4.4 (#1) and 11 (#2) $\times 10^{-2}$ $\Omega$ cm.

## 3. Materials and Methods

Polycrystalline samples of $Li_xWO_3$ ($x$ = 0.5, 0.6, 0.7, 0.8, 0.85, 1.0) were prepared using a high-pressure (HP) technique. Stoichiometric mixtures of $Li_2WO_4$ (99%, Kojundo Chemical, Tokyo, Japan), $WO_2$ (99%, Rare Metallic, Tokyo, Japan), and $WO_3$ (99.999%, Rare Metallic, Tokyo, Japan) were ground in a mortar for 30–60 min and pressed into a pellet. Each pellet was sealed in a platinum capsule, inserted in a graphite tube heater, and enclosed in a pyrophyllite cube. These procedures were carried out in an $N_2$-filled grove box. The pressure applied was 2 GPa for $Li_{0.5}WO_3$ and 5–8 GPa for a $Li_xWO_3$ ($x > 0.5$) using a cubic-anvil press, and the temperature was kept at 850–1200 °C for 30 min during the reaction (see more detail in Table 2.). We observed a tendency that higher pressure ($\geq$5 GPa) gives a higher purity phase.

**Table 2.** Synthesis conditions of $Li_xWO_3$.

| $x$ | 0.5 | 0.6 | 0.7 | 0.8 | 0.85 | 1.0 |
|---|---|---|---|---|---|---|
| **Pressure (GPa)** | 2 | 5 | 5 | 6 | 6 | 8 |
| **Temperature (°C)** | 850 | 850 | 1000 | 1200 | 1200 | 1200 |

Powder XRD patterns of polycrystalline samples of $Li_xWO_3$ were collected using an X-ray powder diffractometer (Bruker D8 Advance diffractometer, Cu $K\alpha$), with the accelerating voltage and the applied current of 40 kV and 40 mA. Diffraction peaks were recorded in the $2\theta$ range of 10–80° with a scan step of 0.2°·s$^{-1}$. Powder neutron diffraction for $Li_{0.8}WO_3$ was collected at room temperature using a high-resolution powder diffractometer BT-1 at the National Institute of Standards and Technology (NIST) Center for Neutron Research (NCNR) (Gaithersburg, MD, USA). A Cu(311) monochromator was used to produce monochromatic neutron beams with a wavelength of 1.5397 Å. The powder sample was loaded into a vanadium cell. Rietveld refinement was performed using the JANA2006 package (Vaclav Petricek, Michal Dusek, and Lukas Palatinus, Prague, Czech Republic). Electrical resistivity of samples was collected using the Physical Properties Measurement System (PPMS, Quantum Design, San Diego, CA, USA) by the four-probe method using Au wire and Ag paste. The experimental setup was done in a glovebox right after the synthesis.

## 4. Conclusions

We have succeeded in expanding the Li concentration in the lithium tungsten bronze $Li_xWO_3$ using the high-pressure synthesis. Unlike the lower Li phase ($0 < x \leq 0.5$) with the perovskite phase ($Im-3$), the new phase adopts the $LiNbO_3$-type structure ($R3c$), notably with a variable A-site composition and thus electron count. The structural change can be regarded as the conversion of octahedral rotations from three in-phase rotations ($a^+a^+a^+$) to three out-of-phase rotations ($a^-a^-a^-$). $Li_{0.8}WO_3$ exhibits a metallic conductivity at low temperature. This study suggests a possibility that other $LiNbO_3$-type compounds could also be susceptible to *A*-site deficiency, which may lead to exotic phenomena.

**Author Contributions:** K.I., Y.I., C.T., H.T., and H.K. conceived and designed the experiments; K.I., Y.I., C.T., and H.T. performed all the experiments; C.M.B. and H.T. conducted ND experiments; K.I., C.T., H.T., and H.K. analyzed data. K.I. and H.K. wrote the paper, with suggestions from other authors.

**Funding:** The work was supported by CREST (JPMJCR1421) and JSPS KAKENHI (JP16H6439, 17H04849).

**Acknowledgments:** Neutron experiments were performed at BT-1 of the NIST Center for Neutron Research. This work was also supported by the Japan Society for the Promotion of Science (JSPS) Core-to-Core Program (A) Advanced Research Networks.

**Conflicts of Interest:** The authors declare no conflict of interest.

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
