# Peer review of "High-Pressure Synthesis of Non-Stoichiometric LixWO3 (0.5 ≤ x ≤ 1.0) with LiNbO3 Structure"

_inorganics, doi:10.3390/inorganics7050063_

Round 1

Reviewer 1 Report

"High Pressure Synthesis of Non-Stoichiometric LixWO3 (0.5 ≤ x ≤ 1.0) with LiNbO3 Structure" is a well written and interesting manuscript. The experiments appear to be correctly executed and the results are clearly exposed and commented.

Some typos are present (i.e. "defficiency", several times in the manuscript), so the Authors should carefully re-read the manuscript.

Additionally, it would be interesting to know the actual (not relative) conductivity values in Fig. 4. Author could add a second scale or a second plot or change the figure altogether.

Author Response

Response to Reviewer 1:

"High Pressure Synthesis of Non-Stoichiometric LixWO3 (0.5 ≤ x ≤ 1.0) with LiNbO3

Structure" is a well written and interesting manuscript. The experiments appear to be

correctly executed and the results are clearly exposed and commented.

Response: We are delighted to know the positive comments to our manuscript.

Some typos are present (i.e. "defficiency", several times in the manuscript), so the Authors

should carefully re-read the manuscript.

Response: The types have been corrected. We carefully checked the manuscript.

Additionally, it would be interesting to know the actual (not relative) conductivity values

in Fig. 4. Author could add a second scale or a second plot or change the figure altogether.

Response: The actual values have been provided in the caption of Figure 4.

Reviewer 2 Report

This paper is suitable for publication after some changes. 

More information about the high pressure preparation is needed. What apparatus was used? What is the large temperature range--uncertainty or different runs? Is it reasonable that the density of the perovskite and LiNbO3 phases are so different? Is that seen in other perovskite to LiNbO3 transitions? Is the phase boundary exactly at 0.5 Li? Only figure 2b shows results versus x. What happens for smaller x--only perovskite, or was not run? With such a large density difference, it should be possible to make LiNbO3 structure for x<0.5 too. Was this tried?

By the way, the authors should also show how they know the sample is not ilmenite structure rather than LiNbO3. If they cannot tell, they should not claim LiNbO3.

Author Response

Response to Reviewer 2:

This paper is suitable for publication after some changes.

Response: We are delighted to know the positive comments to our manuscript.

More information about the high pressure preparation is needed. What apparatus was used?

What is the large temperature range--uncertainty or different runs?

Response: Detailed information on the high-pressure synthesis has been provided

in the revised manuscript, with Table 2 being added.

Is it reasonable that the density of the perovskite and LiNbO3 phases are so different? Is that

seen in other perovskite to LiNbO3 transitions?

Response: We thank the careful reading. We checked our data and found that the

calculation was wrong. There is only a subtle difference (0.40%). We have revised

the corresponding sentence. (Page 2, 1st paragraph).

Is the phase boundary exactly at 0.5 Li? Only figure 2b shows results versus x. What

happens for smaller x--only perovskite, or was not run? With such a large density difference,

it should be possible to make LiNbO3 structure for x<0.5 too. Was this tried?

Response: We appreciate the reviewer for valuable comments. Since we have not

tried the high-pressure reaction for x < 0.5, we cannot tell about the phase boundary.

We plan to do this experiment in future. In the revised manuscript, we addressed

that the LiNbO3 phase may be extended to a lower x.

By the way, the authors should also show how they know the sample is not ilmenite structure

rather than LiNbO3. If they cannot tell, they should not claim LiNbO3.

Response: Ilmenite structure has R−3 symmetry with different reflection conditions

from R3c. For example, the 003 and 101 peaks (2θ = 27.7˚ and 28.6˚) allowed by R−3

are absent in our XRD (Fig. 2(a)), suggesting R3c is likely. We added a sentence

indicating the observed pattern is not compatible with the ilmenite structure (Page

3).

The manuscript reports an interesting study of the LixWO3 crystal chemistry. By exploiting

high pressure synthesis the x-range is extended beyond the known and a LiNbO3-type

structures are obtained, possibly non-centrosymmetric and of interest for studies of

ferroelectrics. Further studies would be of high interest as the present study does not provide

a convincing conclusion about the space group. However, the analysis and the discussion of

the results are generally sound and I have only a few minor suggestion that I would like the

authors to address before I can recommend before publication.

Response: We would like to appreciate the reviewer for his/her positive comment.

Reviewer 3 Report

The manuscript reports an interesting study of the LixWO3 crystal chemistry. By exploiting high pressure synthesis the x-range is extended beyond the known and a LiNbO3-type structures are obtained, possibly non-centrosymmetric and of interest for studies of ferroelectrics. Further studies would be of high interest as the present study does not provide a convincing conclusion about the space group. However, the analysis and the discussion of the results are generally sound and I have only a few minor suggestion that I would like the authors to address before I can recommend before publication.

The introduction is generally well-written, but I miss a few things. You could consider also discussing results in relation to:

1) lithium intercalation into bronze WO3, e.g. 10.1103/PhysRevB.46.2554

2) (Li,Na)WO3, e.g. Li0.51Na0.42WO3 perovskite reported in 10.1021/ja01151a147

I'm curious if any high pressure syntheses were performed for x<0.5 and if this can extend the x-range for LiNbO3-type to lower values.

Why was the x=0.5 synthesis performed with a significantly lower pressure of 2 GPa?

Please specify the pressure used for each synthesis. What do you mean by pressures of 5-8 GPa and why not state the pressure used for each synthesis. Consider adding supporting information with detailed methods and all structural parameters obtained.

How was the pressure determined? What calibration method?

Table 1: several issues:

1) I can guess that "g" is the site occupancy but this is an unusual letter for this parameter and it is not defined.

2) Check units for Uiso. Legend says values are to be multiplied by 100, but the numbers are small

3) the z-coordinate for Li refines to 0.2502(6) for XRD which has little sensitivity regarding this parameter. The uncertainty is remarkably low and suggestive of centrosymmetric R-3c. This is strong contrast the more trustworthy value determined from neutron diffraction, which deviate very significantly from 1/4 (by 15.5 sigma).

It would improve the paper significantly if more neutron diffraction could be performed, multiple sample and different temperature, but I understand the challenges and long delay of publication. Theoretical calculations of stoichiometric LiWO3 could possibly be a quick way of adding more credibility to the (tentative) assignment of space group.

A few minor topographical issues:

- use italics for variables consistently

- page 1 line 71, N2-flled

- page 4 line 134, electroical resistivity

If the authors can address these minor concerns, I think the study should be published.

Author Response

Response to Reviewer 3:

The introduction is generally well-written, but I miss a few things. You could consider also

discussing results in relation to:

1) lithium intercalation into bronze WO3, e.g. 10.1103/PhysRevB.46.2554

2) (Li,Na)WO3, e.g. Li0.51Na0.42WO3 perovskite reported in 10.1021/ja01151a147

Response: The former paper is added as Ref. 11. The latter paper is not included

since it shows co-doped materials, which makes a comparison with our material

difficult.

I'm curious if any high pressure syntheses were performed for x<0.5 and if this can extend

the x-range for LiNbO3-type to lower values.

Response: We appreciate the reviewer for valuable comments. We have not tried

high-pressure synthesis of x < 0.5 and we plan to do the experiment in future (see

the response to Reviewer #2).

Why was the x=0.5 synthesis performed with a significantly lower pressure of 2 GPa? Please

specify the pressure used for each synthesis. What do you mean by pressures of 5-8 GPa and

why not state the pressure used for each synthesis. Consider adding supporting information

with detailed methods and all structural parameters obtained. How was the pressure

determined? What calibration method?

Response: We have added Table 2 that summarizes the synthesis conditions. For

calibrations, we did when installing the high-pressure machines. The pressure was

determined using the resistivity transitions of Bi and Sn, according to Bean, V.E et.

al., Physica 139 & 140B (1986) 52-54. There is no special meaning of employing a

lower pressure of 2 GPa for x = 0.5. It is not shown in our manuscript, but our major

efforts were devoted to identifying a possible superconductivity for a higher x

range of 0.7 ~ 0.9 (which finally turned out to be fake after one-year examination!);

We applied many reaction conditions (temperature, pressure) and observed a

tendency that higher pressure (≥ 5 GPa) is better to obtain a high-purity phase. This

tendency has been described in the revised manuscript (Page 5).

Table 1: several issues:

1) I can guess that "g" is the site occupancy but this is an unusual letter for this parameter

and it is not defined.

2) Check units for Uiso. Legend says values are to be multiplied by 100, but the numbers

are small

Response: Modified as suggested.

3) the z-coordinate for Li refines to 0.2502(6) for XRD which has little sensitivity regarding

this parameter. The uncertainty is remarkably low and suggestive of centrosymmetric R-3c.

This is strong contrast the more trustworthy value determined from neutron diffraction,

which deviate very significantly from 1/4 (by 15.5 sigma).

Response: Although the standard deviation is rather low, we think that the actual

error should be larger. The neutron result is definitely more reliable in terms of zcoordinate

of Li.

It would improve the paper significantly if more neutron diffraction could be performed,

multiple sample and different temperature, but I understand the challenges and long delay

of publication. Theoretical calculations of stoichiometric LiWO3 could possibly be a quick

way of adding more credibility to the (tentative) assignment of space group.

Response: Thank for the reviewer’s suggestion. It would be desirable to include the

DFT calculations, however we would like to do it in future as we prefer earlier

publications.

A few minor topographical issues:

- use italics for variables consistently

- page 1 line 71, N2-flled

- page 4 line 134, electroical resistivity

If the authors can address these minor concerns, I think the study should be published.

Response: All errors were corrected.
